# Prostate-Specific Antigen Bounce after ^125^I Brachytherapy Using Stranded Seeds with Intraoperative Optimization for Prostate Cancer

**DOI:** 10.3390/cancers14194907

**Published:** 2022-10-07

**Authors:** Tae Hyung Kim, Jason Joon Bock Lee, Jaeho Cho

**Affiliations:** 1Yonsei Cancer Center, Department of Radiation Oncology, Yonsei University College of Medicine, Seoul 03722, Korea; 2Department of Radiation Oncology, Nowon Eulji Medical Center, Eulji University School of Medicine, Seoul 01830, Korea; 3Department of Radiation Oncology, Kangbuk Samsung Hospital, Sungkyunkwan University School of Medicine, Seoul 03181, Korea

**Keywords:** prostate cancer, brachytherapy, bounce, prognostic factor

## Abstract

**Simple Summary:**

Our study investigated clinical features of prostate-specific antigen (PSA) bounce in patients undergoing brachytherapy. PSA bounce is common and discriminating between large bounces and biochemical failures is very difficult. Therefore, we suggest important points to discriminate between large bounces and biochemical failures. In addition, we aimed to examine the clinical features and details of PSA bounce in patients receiving brachytherapy.

**Abstract:**

Prostate-specific antigen (PSA) bounce is common in patients undergoing ^125^I brachytherapy (BT), and our study investigated its clinical features. A total of 100 patients who underwent BT were analyzed. PSA bounce and large bounce were defined as an increase of ≥0.2 and ≥2.0 ng/mL above the initial PSA nadir, respectively, with a subsequent decline without treatment. Biochemical failure was defined using the Phoenix definition (nadir +2 ng/mL), except for a large bounce. With a median follow-up of 49 months, 45% and 7% of the patients experienced bounce and large bounce, respectively. The median time to bounce was 24 months, and the median PSA value at the bounce spike was 1.62 ng/mL, a median raise of 0.44 ng/mL compared to the pre-bounce nadir. The median time to bounce recovery was 4 months. The post-bounce nadir was obtained at a median of 36 months after low-dose-rate BT. On univariate analysis, age, the PSA nadir value at 2 years, and prostate volume were significant factors for PSA bounce. The PSA nadir value at 2 years remained significant in multivariate analysis. We should carefully monitor young patients with high prostate volume having a >0.5 PSA nadir value at 2 years for PSA bounce.

## 1. Introduction

Several radiotherapeutic approaches are available for the definitive treatment of prostate cancer, including low-dose-rate (LDR) and high-dose-rate brachytherapy (BT), external beam radiotherapy (EBRT) with intensity-modulated RT, proton therapy, and even carbon ion therapy [1]. LDR-BT is a well-established standard treatment for early prostate cancer and offers excellent oncological outcomes, dosimetric advantages, and patient convenience [2,3,4].

Prostate-specific antigen (PSA) is a sensitive diagnostic and prognostic marker of prostate cancer. PSA bounce, a temporary increase in PSA levels and a subsequent decrease without intervention, occurs in 15–84% of men receiving ^125^I BT [5]. Several studies and trials have demonstrated that PSA bounce after ^125^I BT is a good prognostic factor [5,6], and young age is generally accepted as a predictive factor for PSA bounce [7,8]. In addition, few studies have investigated the difference between biochemical failure and PSA bounce [9,10,11], and most of them were single-center studies with several limitations. The predictive factors for PSA, as well as PSA details, remain unclear.

Therefore, this investigation aimed to examine the clinical features and details of PSA bounce in patients receiving LDR-BT.

## 2. Materials and Methods

### 2.1. Patients

From November 2012 to December 2017, 105 patients with prostate cancer underwent LDR-BT at Yonsei Cancer Center. Risk groups were defined according to the Memorial Sloan Kettering Cancer Center (MSKCC) criteria [12,13] and the D’Amico criteria [14]. Patients with localized and locally advanced disease were treated with LDR-BT, while those with metastatic disease were not. According to MSKCC risk grouping, patients with low risk and patients with intermediate to high risk whose adverse pathologic features approached that of low-risk patients were treated with LDR-BT. Patients treated with androgen deprivation therapy (ADT) before LDR-BT were included for analysis, but those who used maintenance ADT were excluded. The date of LDR-BT was day 0 of follow-up. PSA levels were measured prior to LDR-BT, every 3 months for the first year, every 6 months for 3 years, and every 12 months thereafter. Patients with a PSA follow-up duration of <2 years were excluded from the study. After exclusion, the data from 100 patients were analyzed. The procedures followed in this retrospective study were in accordance with the guidelines of the Helsinki Declaration of 1975, revised in 2000, and the study was approved by the Severance Hospital institutional review board (IRB # 4-2019-0767). Because this study was retrospective, the need for written informed consent was waived.

### 2.2. LDR-Brachytherapy

Preoperative treatment planning was performed based on trans-rectal ultrasonography (TRUS) images acquired during preoperative simulation. Radiation oncologist confirmed Pubic arch interference that prevents proper needle insertion into the peripheral zone of the prostate. The prostate and organs at-risk, including the urethra, bladder, rectum, and seminal vesicle, were contoured using the VariSeed software Ver. 8.0.1 (Varian Medical Systems Inc., Palo Alto, CA, USA). The prescribed dose for the prostate was 145 Gy. All patients underwent BT using stranded seeds. Post-implant computed tomography scanning was performed, and post-implant dosimetric evaluation was performed on days 0 and 30 after seed implantation. The dosimetric parameters analyzed in this study were the dose (Gy) received by 90% of the prostate gland (D90), percentage of the prostate volume receiving 100% and 150% of the prescribed peripheral dose (V100/150), and the dose (Gy) received by 90% of the urethra (D90).

### 2.3. Definitions of PSA Bounce

The nadir was defined as the lowest PSA value observed during the entire follow-up period after LDR-BT. The pre-bounce nadir was defined as the lowest PSA value before the bounce [15]. PSA bounce, in this study, was defined as an increase of ≥0.2 ng/mL above the pre-bounce nadir, with a subsequent decline without treatment [5,6]. The time before bounce was defined as the time elapsed between LDR-BT and the first PSA bounce. The bounce duration was defined as the elapsed time between the first PSA bounce and the PSA value that is less than the pre-bounce nadir. The bounce magnitude was defined as the difference in the PSA value between the pre-bounce nadir and bounce. A large PSA bounce was defined as a PSA increase ≥2.0 ng/mL above the nadir, with a subsequent decline to or below the initial nadir without treatment. Biochemical failure after LDR-BT was defined using the American Society for Therapeutic Radiology and Oncology Phoenix definition (nadir +2 ng/mL), except for a large PSA bounce [16].

To discriminate a large PSA bounce and biochemical failure, careful history taking was performed to determine if the condition was transient prostatitis. If acute prostatitis was suspected, anti-bacterial and anti-inflammatory drugs were administered. In addition, one or two more close follow-up with imaging studies to confirm recurrence was performed. If no signs of recurrence were present on imaging, no salvage treatment such as hormone therapy was initiated. Subsequent decrease in PSA value was considered as large PSA bounce.

PSA value decreases dramatically after initiation of ADT; therefore, its discontinuation can result in PSA elevation, which we defined as hormone withdrawal rebound. PSA bounce was determined after the PSA value showed a downward trend for patients who were treated with ADT.

### 2.4. Statistical Analysis

Statistical analyses were conducted using SPSS version 25.0 (IBM Corp., Armonk, NY, USA). Differences in characteristics and toxicities were compared using chi-square tests. Logistic regression modeling was performed for univariate and multivariate analyses to identify predictive factors for PSA bounce and large PSA bounce. Factors showing *p* < 0.10 in the univariate analysis were included in the multivariate analysis. Statistical significance was defined as *p* < 0.05. To assess the cutoff point of the PSA nadir value for predicting PSA bounce, receiver operating characteristic curve analysis was used and the area under the curve (AUC) was also calculated.

## 3. Results

### 3.1. Patient Characteristics

The patient characteristics are shown in Table 1. The median age was 64 years (interquartile range [IQR] 58.5–70). Most patients had Gleason 3 + 3 (55%), and 21%, 13%, and 11% of patients had Gleason 3 + 4, 4 + 3, and 4 + 4, respectively. PSA was greater than 20 ng/mL in 4% of the patients and greater than 10 ng/mL in 19%; altogether, 45%, 49%, and 6% of patients were in the low, intermediate, and high-risk groups, respectively, as per the MSKCC criteria. Seventeen patients underwent ADT before LDR-BT. Seven patients received ADT because of large prostate volume, five because of high Gleason score, and five because of delay in LDR-BT. Before LDR-BT, the mean prostate volume was 28.9 cc, and the median D90 at post-implant 0 day was 149.9 Gy. Patients who experienced PSA bounce were young (*p* = 0.005), had a large prostate volume (*p* = 0.024), and had a greater number of implanted seeds (*p* = 0.03) than those who did not.

### 3.2. Analysis of Bounce Phenomenon

The median follow-up period was 49 months (24–100). Among 17 patients who underwent ADT before LDR-BT, 8 patients (47%) experienced hormone withdrawal rebound. PSA bounce occurred in 45 patients (45%), and a large PSA bounce occurred in 7 patients (7%). Figure 1 shows the PSA changes for all patients, and the PSA values of the bounce population fluctuated dynamically. In the bounce population, the median PSA value at pre-bounce nadir was 0.92 ng/mL (IQR, 0.49–1.82), and the median time to pre-bounce nadir was 10 months (IQR, 7–16). The median time to bounce was 24 months (IQR, 16–29) after LDR-BT. The median PSA value at the bounce spike was 1.62 ng/mL (IQR, 0.94–2.61), corresponding to a median raise of 0.44 ng/mL (IQR range, 0.29–0.83) compared to the pre-bounce nadir (Figure 2). The median time from PSA bounce to the date of bounce recovery was 4 months (IQR, 3–8; Figure 3). Twenty patients (46%) with a PSA bounce had a decreased PSA level within 6 months (Figure 4). The median PSA value at post-bounce nadir was 0.36 ng/mL (IQR, 0.20–0.91), obtained at a median of 36 months (IQR, 26–45) after LDR-BT.

In the no-bounce population, the median PSA value at nadir was 0.22 ng/mL (IQR, 0.12–0.37), obtained in a median of 28 months (IQR, 20–38). There was no significant difference in the nadir values between the bounce and no-bounce populations (*p* = 0.061). The median time to obtain the nadir was significantly lower in the no-bounce population than in the bounce population (*p* = 0.024).

On univariate analysis, age at LDR-BT (*p* = 0.007), PSA nadir value at 2 years (*p* < 0.001), and prostate volume before LDR-BT (*p* = 0.022) differed significantly between the entire cohort and patients with PSA bounce. Odds ratios (ORs) were 0.928 for age at LDR-BT (95% confidence interval (CI) 0.879–0.980), 3.873 for PSA nadir value at 2 years (95% CI 1.817–8.254), and 1.064 for prostate volume before LDR-BT (95% CI 1.009–1.123). On multivariate analysis, the PSA nadir value at 2 years was the most powerful predictor of PSA bounce (*p* = 0.014; Table 2). The cut-off of PSA nadir value at 2 years was 0.5 (AUC 0.732). The rate of PSA bounce was significantly lower in patients whose PSA nadir value at 2 years was below 0.5 ng/mL (24% vs. 71%, *p* < 0.001) than in whose PSA nadir values were >0.5 ng/mL.

### 3.3. Analysis of Large Bounce Phenomenon

Seven patients had large PSA bounces. In the large bounce population, the median PSA value at pre-bounce nadir was 1.91 ng/mL (IQR, 0.44–3.29), and the median time to pre-bounce nadir was 11 months (IQR, 7–12). The median time to large bounce was 24 months (IQR, 16–29) after LDR-BT. The median PSA value at the bounce spike was 4.84 ng/mL (IQR, 4.30–6.21), corresponding to a median raise of 2.72 ng/mL (IQR range, 2.36–3.80) compared to the pre-bounce nadir. The median time from large PSA bounce to the date of bounce recovery was 2 months. Five patients (71%) had a decreased PSA level within 3 months, and 2 patients had decreased PSA levels at 13 months and 14 months, respectively.

Univariate analysis showed that the age at LDR-BT (OR, 0.759; 95% CI, 0.640–0.900; *p* = 0.001) and PSA nadir value at 2 years (OR, 4.008; 95% CI, 1.640–9.797; *p* = 0.002) differed significantly between the whole cohort and patients with large PSA bounce. In multivariate analysis, age and the PSA nadir value at 2 years were significant prognostic factors (*p* = 0.005 and 0.029, respectively, as shown in Table 3.

### 3.4. Early Clinical Outcomes

No intraprostatic gross failure or prostate cancer-related death was reported, but one regional failure and two biochemical failures were noted. One patient died 24 months after LDR-BT because of adenocarcinoma of the right upper lung, pathologically confirmed lung cancer, and non-metastatic prostate cancer. All failure patients had Gleason 4 + 3 or 4 + 4, whereas five patients (71%) with large PSA bounces had Gleason 3 + 3. Other clinical or dosimetric characteristics were similar between patients with failure and those with large PSA bounces. The median age of failure patients and large PSA bounce patients was 64 and 52 years (*p* = 0.008), respectively.

Two patients experienced biochemical failure at 43 and 31 months after LDR-BT. Both patients had a Gleason score of 7 (4 + 3), PSA values lower than 10 ng/mL, and clinical T2a stage. Salvage ADT was administered, and a PSA value below the nadir was achieved. One patient developed lymph node metastases. The left obturator lymph node was found 48 months after LDR-BT and treated with hypofractionated EBRT of 45 Gy in five fractions, and salvage ADT was used. After EBRT, the PSA value decreased to the undetected range, and the lymph node disappeared radiographically.

## 4. Discussion

In this study, with a median follow-up time of 49 months, 45%, 7%, and 3% of the patients experienced PSA bounce, large PSA bounce, and biochemical failure, respectively. Young patients (<60 years), whose PSA nadir values were >0.05 at 2 years and had large prostate volumes (>30 cc) had a high probability of having PSA bounce. Our study also showed no difference in nadir values between the bounce and no-bounce populations. The median time to obtain the nadir in the no-bounce population was significantly lower than that of the bounce population. In addition, we suggest important points to discriminate between large bounces and biochemical failures.

After successful surgery, the patient’s PSA level should rapidly decrease to undetectable levels. However, BT or EBRT may take up to 5 years after treatment to achieve a final nadir in PSA. This is because of the slower tumor cell-killing process with RT, resulting in a gradual decrease in PSA [17]. Several definitions of PSA bounce have previously been used. These include an increase of ≥0.1 [18], ≥0.2 [19], 0.4 [20], 0.5 ng/mL [21], or simply an increase of any magnitude [22]. However, a PSA rise of ≥0.2 ng/mL is the most frequently used definition among those mentioned previously. One of the reasons is the need for a definition that minimizes ‘‘noise’’ due to laboratory testing errors. In addition, because a bounce defined as a rise of ≥0.2 ng/mL has been used by most previous publications, it allows for comparison among reports. Our study used the definition of ≥0.2 ng/mL, and the incidence of bounce was 45% in patients who received LDR-BT.

Young patients, usually younger than 65 years, experience a PSA bounce more often. A controlled study was conducted to confirm the hypothesis that a higher frequency of sexual intercourse in the young population causes a higher rate of PSA increase than in older patients [23]. Since ejaculation has been associated with transient elevation of PSA [24], a questionnaire about sexual function was administered. However, no between-group differences in sexual function were observed. An immune reaction can explain PSA bounce. Patients who experienced a PSA bounce had a higher density of cluster of differentiation (CD) 3 and CD8 lymphocyte populations within the tumor, assessed by blood samples [25]. The authors suggested that the strength of the immune response decreases with age, which can explain the decreased bounce rates in the older adult population. Furthermore, this immunologic reaction could have a systemic effect on metastasis, explaining the decrease in biological relapses and improved overall survival rate seen in the bounce population [26].

It is generally accepted that a larger prostate volume results in more frequent PSA bounces. In our study, patients with large prostate volumes had more frequent PSA bounces. According to Stock et al. [20], patients with larger prostate volumes had a 23% increased risk of bounce at 5 years after treatment. In a study by Merrick et al. [27], the transition zone volume was a predictive factor for PSA bounce but not prostate volume. The investigators stated that the increased risk of bounce might be related to the possibility that benign prostatic elements, such as benign prostatic hyperplasia, could respond to RT with different PSA kinetics than malignant cells.

When patients whose PSA nadir values increased over 2.0 ng/mL after BT, distinguishing biochemical failure and large PSA bounce is most difficult situation encountered in clinical practice because they use the same value to discriminate. However, few studies have investigated large-magnitude bounce and predictive factors for large PSA bounce. Several studies concluded that patients who experienced large PSA bounce were significantly younger than those with biochemical failure [11,28]. Herein, we suggest follow-up protocol for patients whose PSA nadir values increase over 2.0 ng/mL after BT. First, there is the possibility of transient acute prostatitis. Careful history taking and physical examination should be performed for differential diagnosis. Next, we recommend identifying recurrence through an imaging study. Finally, time of occurrence of PSA level increase could be the discrimination point between biochemical failure and large PSA bounce. PSA bounce occurs within 30 months after treatment; on the contrary, biochemical failure often occurs after 30 months post-treatment [29]. In addition, as mentioned above, younger age and PSA nadir values could be a predictive factor for large PSA bounce to ensure accurate estimation of treatment efficacy and avoid unnecessary salvage treatment.

Several studies investigated the prognostic impact of PSA bounce and found increased freedom from biochemical failure or prolonged disease specific and overall survival [26,29]. There are several reports insisting that PSA bounce after EBRT is a factor for poor prognosis [30]. In our study, the number of recurrences is too small to analyze the relation between PSA bounce and biochemical failure. However, the relation between PSA bounce and biochemical failure is debatable, and this relation will be studied by a long follow-up period with more patients treated with LDR-BT.

This study had several limitations. First, it was an institutional-based retrospective study, which introduced potential biases. Second, this study had a small sample size, which had a statistically lower power, and the follow-up period might have been short. In addition, the number of recurrences is too small to analyze the association between PSA bounce and biochemical failure. However, our study included data from treatment with the same protocol and evaluation with the same dosimetric parameters for all patients. Despite these limitations, we present lessons for follow-up patients who received LDR-BT. Patients who had PSA values lower than 0.05 2 years after LDR-BT had a low probability of PSA bounce. Furthermore, discrimination between a PSA bounce and a biochemical failure could be possible based on the time of the PSA rise after LDR-BT; a bounce occurred within 2 years, but a biochemical failure occurred after 2 years. Most bounces resolved within 6 months.

## 5. Conclusions

In conclusion, age at LDR-BT was a significant predictive factor for PSA bounce and a large PSA bounce. Patients with a large prostate volume before LDR-BT tended to have a PSA bounce. Therefore, we should carefully monitor patients who are young, have a high prostate volume, and have PSA nadir values more than 0.5 at 2 years for the possibility of PSA bounce.

## Figures and Tables

**Figure 1 cancers-14-04907-f001:**
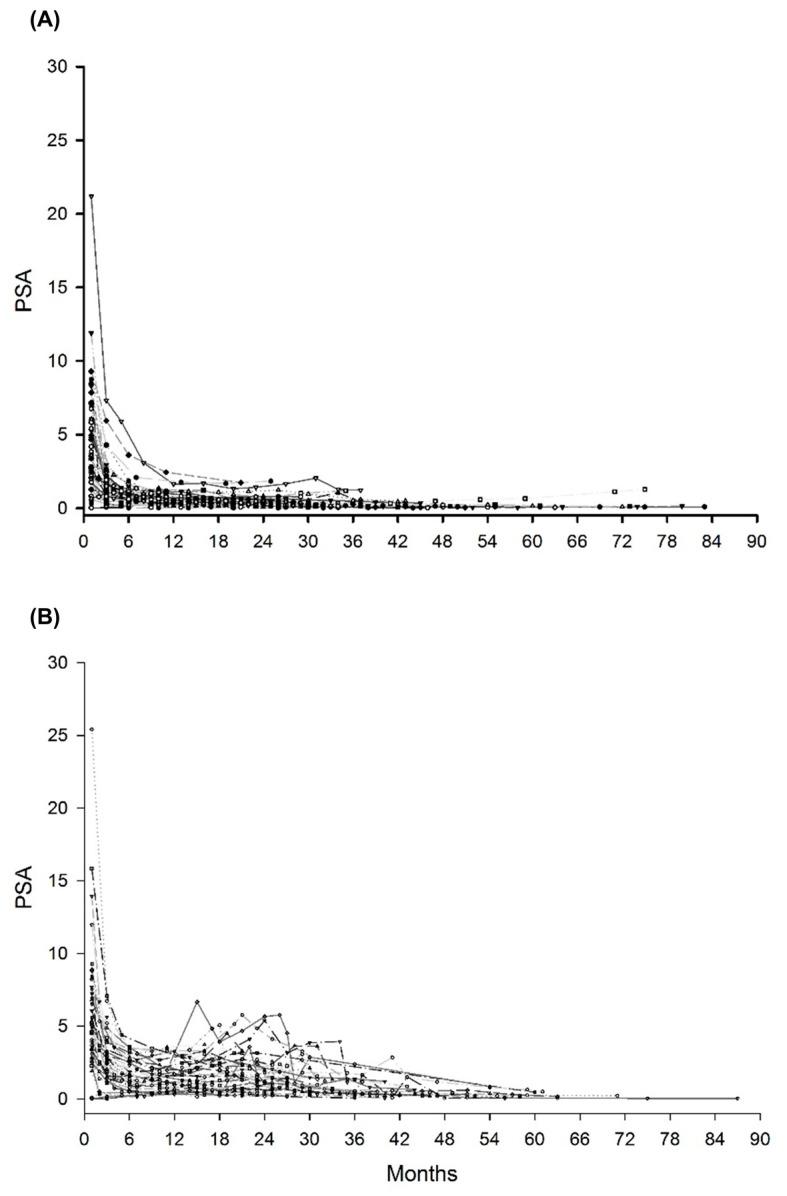
(**A**) PSA value for patients without bounce; (**B**) PSA value for patients with bounce; (**C**) PSA value for patients with large bounce; (**D**) PSA value for patients with failure. PSA, prostate-specific antigen.

**Figure 2 cancers-14-04907-f002:**
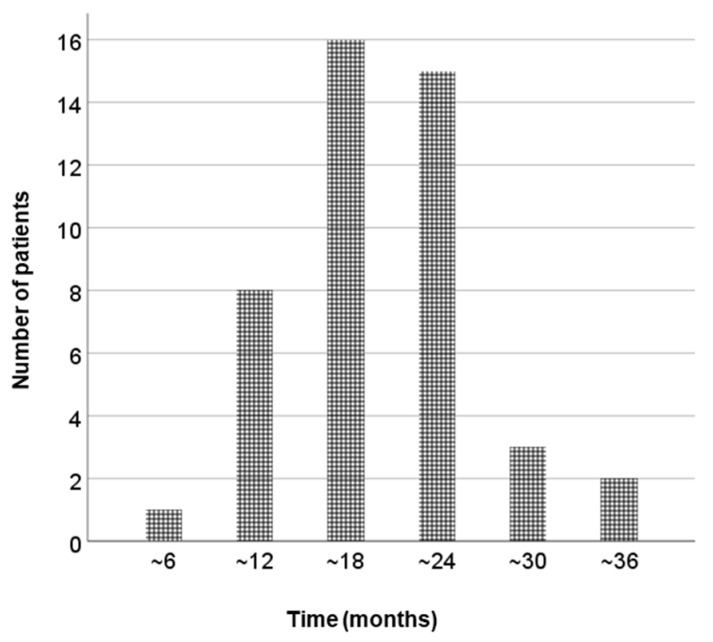
The period before PSA bounce. PSA, prostate-specific antigen.

**Figure 3 cancers-14-04907-f003:**
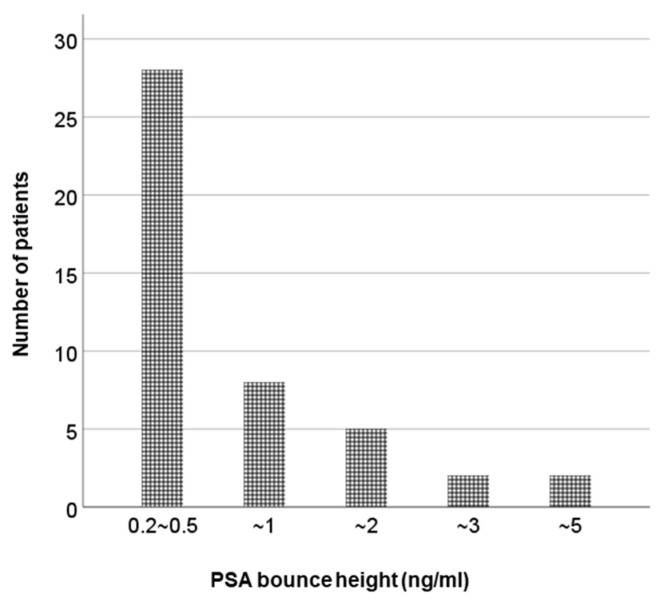
The PSA bounce magnitude. PSA, prostate-specific antigen.

**Figure 4 cancers-14-04907-f004:**
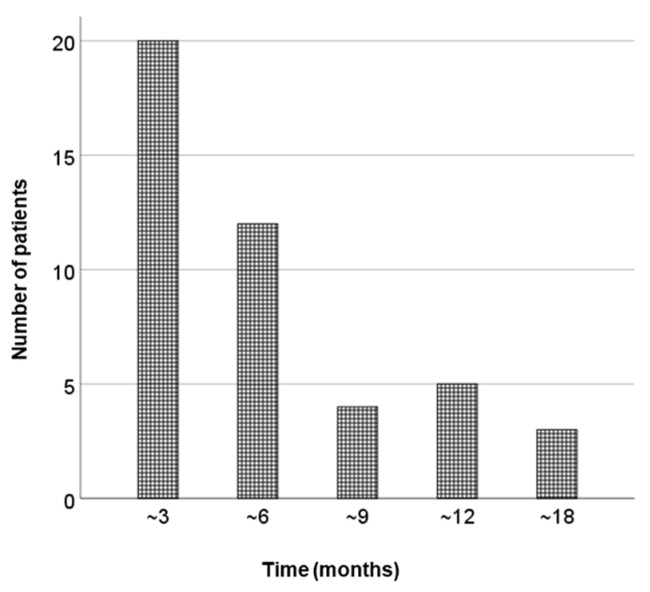
Time to recover from PSA bounce. PSA, prostate-specific antigen.

**Table 1 cancers-14-04907-t001:** Patients’ characteristics.

Characteristic	Total	No Bounce	PSA Bounce	*p*-Value
	n = 100	n = 55	n = 45	
Age (years)				0.005
Range	46–82	50–82	46–77	
Median (Q1–Q3)	64 (59–70)	65 (61–73)	60 (57.5–68)	
Gleason score				0.411
6 (3 + 3)	55 (55%)	28 (50%)	27 (60%)	
7 (3 + 4)	21 (21%)	11 (20%)	10 (22%)	
7 (4 + 3)	13 (13%)	8 (15%)	5 (11%)	
8 (4 + 4)	11 (11%)	8 (15%)	3 (7%)	
T stage				0.512
T1c-T2a	79 (79%)	43 (78%)	36 (80%)	
T2b-T2c	21 (21%)	12 (22%)	9 (20%)	
Pre-BT PSA value, ng/mL				0.245
Range	2.8–32.9	3.0–20.6	2.8–32.9	
Median (Q1–Q3)	7.4 (5.5–9.7)	7.4 (5.6–9.3)	7.4 (5.4–11.9)	
Pre-BT PSA, n (%)				0.440
<10	77 (77%)	44 (80%)	33 (73%)	
10–20	19 (19%)	10 (18%)	9 (20%)	
≥20	4 (4%)	1 (2%)	3 (7%)	
MSKCC risk group [12]				0.815
Low	45 (45%)	25 (46%)	20 (44%)	
Intermediate	49 (49%)	26 (47%)	23 (51%)	
High	6 (6%)	4 (7%)	2 (4%)	
D’Amico risk group [14]				0.941
Low	37 (37%)	20 (36%)	17 (38%)	
Intermediate	46 (46%)	25 (46%)	21 (47%)	
High	17 (17%)	10 (18%)	7 (15%)	
ADT before BT				0.728
No	83 (83%)	45 (82%)	38 (84%)	
Yes	17 (17%)	10 (18%)	7 (16%)	
Pre-BT prostate volume, cc				0.024
Range	14.0–48.0	16.7–44.7	14.0–48.0	
Median (Q1, Q3)	28.9 (23.8–35.7)	27.1 (22.9–32.8)	31.2 (24.3–39.3)	
Number of implanted seeds				0.030
Median (Range)	76 (52–102)	74 (52–100)	80 (55–102)	
D90, Gy, median (range)	149.9 (131.0–174.9)	151.1 (131.1–173.5)	149.5 (131.0–174.9)	0.501
PSA, prostate-specific antigen; BT, brachytherapy; ADT, androgen deprivation therapy; MSKCC, Memorial Sloan Kettering Cancer Center				

**Table 2 cancers-14-04907-t002:** The prognostic factors associated with PSA bounce.

Characteristic	Univariate Analysis			Multivariate Analysis		
	HR	95% CI	*p*-Value	HR	95% CI	*p*-Value
Age (≤60 vs. >60 years)	0.296	0.126–0.895	0.005	0.455	0.175–1.183	0.106
Gleason score (6 vs. >6)	0.691	0.312–1.534	0.364			
T stage (T1c and T2a vs. T2b and T2c)	0.896	0.339–2.366	0.824			
Pretreatment PSA value	1.052	0.968–1.143	0.230			
PSA nadir value at 2 years	3.873	1.817–8.254	<0.001	2.657	1.194–5.914	0.017
MSKCC risk group [12] (low vs. intermediate and high)	1.042	0.472–2.300	0.920			
D’Amico risk group [14] (low vs. intermediate and high)	0.941	0.416–2.127	0.884			
Hormone therapy (no vs. yes)	0.829	0.288–2.388	0.728			
Prostate volume (cc) (≤30 vs. >30)	3.083	1.358–7.003	0.007	1.940	0.772–4.875	0.106
PSA, Prostate-specific antigen; MSKCC, Memorial Sloan Kettering Cancer Center						

**Table 3 cancers-14-04907-t003:** The prognostic factors associated with large PSA bounce.

Characteristic	Univariate Analysis			Multivariate Analysis		
	HR	95% CI	*p*-Value	HR	95% CI	*p*-Value
Age (≤60 vs. >60 years)	0.058	0.007–0.506	0.010	0.055	0.005–0.666	0.023
Gleason score (6 vs. >6)	0.465	0.084–2.520	0.375			
T stage (T1c and T2a vs. T2b and T2c)	1.558	0.280–8.661	0.613			
Pretreatment PSA value	1.076	0.956–1.212	0.224			
PSA nadir value at 2 years	4.008	1.640–9.797	0.002	4.961	1.448–16.998	0.011
MSKCC Risk group [12] (low vs. intermediate and high)	0.302	0.056–1.637	0.165			
D’Amico Risk group [14] (low vs. intermediate and high)	1.509	0.278–8.198	0.634			
Hormone therapy (no vs. yes)	2.080	0.369–11.738	0.407			
Prostate volume (cc) (≤30 vs. >30)	3.312	0.611–17.960	0.165			
PSA, prostate-specific antigen; MSKCC, Memorial Sloan Kettering Cancer Center						

## Data Availability

The datasets generated and/or analyzed during the current study are not publicly available due to risk of personal information leakage but are available from the corresponding author on reasonable request.

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
