# Peer review of "Prostate-Specific Antigen Bounce after 125I Brachytherapy Using Stranded Seeds with Intraoperative Optimization for Prostate Cancer"

_cancers, 2022, doi:10.3390/cancers14194907_

Round 1

Reviewer 1 Report

In the manuscript entitled “Prostate specific Antigen Bounce after 125I Brachytherapy Using Stranded Seeds With Intraoperative optimization for Prostate cancer” by Kim et al, the authors have investigated the clinical features of PSA bounce in patients who have undergone brachytherapy and have tried to point out a few important features that discriminate large PSA bounce from biochemical failures. This is i an important study.  However, the following points need further clarification:

The major focus of this work is to determine certain factors/features that discriminate PSA bounce from biochemical failures.   Authors have indicated that PSA bounce is ≥0.2 ng/ml and large bounce is ≥2.0 ng/ml above the initial PSA nadir value. However it is not clear how the difference between biochemical failure and large PSA bounce was determined.

It is not clear whether these patients have localized prostate cancer or have advanced or metastatic disease. Also, among the recipients of LDR-BT, a subset of patients had received ADT. It not clear how  the effect of  ADT was taken into account as reports suggest that ADT has significant impact on PSA bounce as well.

Authors have concluded that patients who experienced PSA bounce were young and also had large prostate volumes. However, the age groups and patient categorizations are not clear.

It is not clear whether PSA bounce can predict biochemical failure in prostate cancer patients. Is there any correlation between PSA bounce or large bounce and biochemical failures in prostate cancer?  Studies have indicated that patients treated with external beam radiation therapy show post treatment PSA bounce and have increased risk of biochemical failure.

Is there any relation between PSA bounce with age, Gleason score and other forms of treatment? Authors have mentioned age at low dose brachytherapy as significant contributing factor for PSA bounce as well as large PSA bounce. Also, patients with large prostate volume have more chances of PSA bounce. However, the contribution of these factors are not very clear.   

Reviewer 2 Report

In this study, Kim and colleagues comprehensively examined the clinical characterization of PSA bounce in patients receiving brachytherapy. Based on the analysis of clinical data from 100 patients, they discovered that age at LDR-BT was a significant predictive factor for large PSA bounce. Furthermore, prostate volume is also a predictive factor for PSA bounce. Therefore, these results suggest that the young patients with large tumor volume and PSA nadir value more than 0.5 may have highest risk for PSA bounce. 

Overall, I think this is a solid clinical research focus on answering a simple but important clinical question. The overall statistic design is sound and clear and results presentation is clear as well. However, I do have several suggestions, which may benefit the manuscript:

1. In table-1, PSA bounce seems correlated with tumor T stage with a p-value of 0.035. However, in table-2, neither gleason score nor T stage has significant correlation with PSA bounce (p=0.364, 0.824). Is there any explanation for this?

2. It is quite interesting that hormone therapy treatment is not related to PSA bounce (p=0.728), which is a little bit surprising as pre-treatment of hormone therapy likely would change the AR dependency of PCa. Is there any explanation of this? 

3. some of the figures have very low resolution and hard to read, such as figure 1.

4. some of the sentence are hard to understand, may need some additional language editing. 
